# Frequency, Etiology, Mortality, Cost, and Prevention of Respiratory Tract Infections—Prospective, One Center Study

**DOI:** 10.3390/jcm11133764

**Published:** 2022-06-29

**Authors:** Wieslawa Duszynska, Marta Idziak, Klaudia Smardz, Anna Burkot, Malgorzata Grotowska, Stanislaw Rojek

**Affiliations:** 1Department and Clinic of Anaesthesiology and Intensive Therapy, Wroclaw Medical University, L. Pasteura Street 1, 50-367 Wroclaw, Poland; idmarta@interia.pl (M.I.); malgorzata.grotowska@umed.wroc.pl (M.G.); 2The Students Scientific Association by Department and Clinic of Anaesthesiology and Intensive Therapy, Wroclaw Medical University, L. Pasteura Street 1, 50-367 Wroclaw, Poland; smardzklaudia@gmail.com (K.S.); annaburkot@wp.pl (A.B.); 3Department of Anaesthesiology and Intensive Therapy, Specialist Hospital in Walbrzych, A. Sokolowskiego Street 4, 58-309 Walbrzych, Poland; staszek.rojek@gmail.com

**Keywords:** respiratory tract infections, ventilator associated pneumonia, non ventilator hospital acquired pneumonia, ventilator associated tracheobronchitis, length of stay, mortality, cost, intensive care unit

## Abstract

Background: Ventilator-associated pneumonia (VAP) is the most monitored form of respiratory tract infections (RTIs). A small number of epidemiological studies have monitored community-acquired pneumonia (CAP), non-ventilator hospital-acquired pneumonia (NV-HAP) and ventilator-associated tracheobronchitis (VAT) in intensive care units (ICUs). The objective of this study was to assess the frequency, etiology, mortality, and additional costs of RTIs. Methods: One-year prospective RTI surveillance at a 30-bed ICU. The study assessed the rates and microbiological profiles of CAP, VAP, NV-HAP, VAT, and VAP prevention factors, the impact of VAP and NV-HAP on the length of ICU stays, and the additional costs of RTI treatment and mortality. Results: Among 578 patients, RTIs were found in 30%. The CAP, NV-HAP, VAP, and VAT rates/100 admissions were 5.9, 9.0, 8.65, and 6.05, respectively. The VAP incidence density/1000 MV-days was 10.8. The most common pathogen of RTI was *Acinetobacter baumannii* MDR. ICU stays were extended by VAP and NV-HAP for 17.8 and 3.7 days, respectively, and these RTIs increased the cost of therapy by 13,029 and 2708 EUR per patient, respectively. The mortality rate was higher by 11.55% in patients with VAP than those without device-associated and healthcare-associated infections (*p* = 0.0861). Conclusions: RTIs are a serious epidemiological problem in patients who are admitted and treated in ICU, as they may affect one-third of patients. Hospital-acquired RTIs extend hospitalization time, increase the cost of treatment, and worsen outcomes.

## 1. Introduction

In terms of epidemiology, respiratory tract infections (RTIs) have been the dominant clinical form of infections, requiring hospital treatment for many years and affect 24–50% of ICU patients [1,2]. Regardless of the definition adopted, when it comes to monitoring nosocomial infections in the ICU, ventilator-associated events (VAE), ventilator-associated pneumonia (VAP), and intubation-associated pneumonia (IAP) most often concern RTIs in mechanically ventilated patients [3,4]. A small number of ICU studies included monitoring community acquired pneumonia (CAP) and nosocomial pneumonia in patients who did not require mechanical ventilation (NV-HAP) as well as lower respiratory tract infections (without symptoms of pneumonia) in mechanically ventilated patients (VAT) [5,6,7,8]. Literature data indicate that VAP/IAP and VAE are the dominant clinical forms of nosocomial infections in the ICU [9,10,11]. VAP frequency increases with more than 60% use of mechanical ventilation (V-UR) [12]. ICU-acquired pneumonia has been shown to increase the risk of mortality in a multicenter, international study [8,13]. Moreover, it has been found, both in highly developed and low-budget countries, that the presence of VAP and NV-HAP extends the duration of ICU treatment and generates additional costs of therapy [3,4,14]. Additionally, NV-HAP was found more frequently than VAP, had similar mortality, and generated higher therapy costs than VAP [15]. The microbial factors for pneumonia may vary by geographic region, country, and even hospital [3,16,17]. The most common VAP pathogens in European ICUs were *Pseudomonas aeruginosa*, *Staphylococcus aureus,* and *Klebsiella pneumoniae*, while in Poland, the dominant strain was *Acinetobacter baumannii* [16,18,19]. In contrast, the most frequent VAP pathogens in American ICUs were *Staphylococcus aureus*, *Pseudomonas aeruginosa*, and *Klebsiella pneumoniae* [17]. A reduction in the incidence of pneumonia in mechanically ventilated patients was found after the implementation of prophylaxis and monitoring standards [2,20,21,22]. The objective of the study was to assess the frequency, etiology, mortality, and additional therapy costs of RTIs as well as compliance with the VAP preventive bundle. 

## 2. Material and Methods

### 2.1. Design and Settings

A prospective observational study was conducted in the period from 1 January 2018 to 31 December 2018 at the Department of Anesthesiology and Intensive Therapy of the Medical University of Wroclaw at a 30-bed ICU. Study involved all the patients hospitalized at ICU during the study period. RTIs were found during routine monitoring of infections. Monthly ICU infection reports from the Infection Monitoring and Treatment Laboratory of the Department of Anesthesiology and Intensive Therapy were used for data collection. The main aim of the study was assessment the frequency of different clinical forms of RTIs in ICU patients: rate of RTIs on admission and during ICU stay, in surgical and medical patients, in terms of gender and age, incidence rates of VAP/NV-HAP/1000 patient-days, and incidence density of VAP/1000 ventilator days. The next elements of the study were the assessment of microbiological factors of infections, compliance with the elements of the VAP prevention package called “VAP-bundles”, and the impact of VAP/NV-HAP on length of stay (LOS) in the ICU, cost of treatment, and mortality.

### 2.2. Ethical Approval

The study was approved by the Bioethics Committee of the Medical University of Wroclaw, No. KB-605/2016, which also included the consent to publish data collected in an anonymous manner. Because all the patients’ data used in this study (including microbiology results) were obtained during routine patient care and monitoring of infections, and a statement covering patients’ data confidentiality was fully respected during data collection and manuscript preparation, no patients’ written consent and statement was needed, according to Bioethics Committee of the Medical University. 

### 2.3. Clinical Diagnosis of RTIs

RTIs were diagnosed based on the criteria of ECDC (European Center for Disease Control and Prevention) and the ENIRRIs project (European Network for ICU-Related Respiratory Infections) [23,24]. Hospital-acquired pneumonia analyzed in the study began in the ICU when it was diagnosed ≥48 h after admission or in other hospital wards when it was diagnosed before or on the day of admission to the ICU. VAP and VAT were diagnosed in a mechanically ventilated patient with an endotracheal tube, whose symptoms appeared after at least 2–3 days of ventilation and admission to the ICU. NV-HAP was diagnosed in a patient who was not mechanically ventilated on the basis of chest X-ray changes, purulent sputum, auscultation changes, increased body temperature > 38 °C, WBC > 12,000/mm^3^, and appropriate microbiological tests (culture of sputum, blood, pleural fluid, smear from the pharynx or bronchial secretions collected immediately after intubating the bronchial tree with a tracheobronchial tube). 

### 2.4. Microbiological Diagnosis of RTI

VAP was diagnosed on the basis of the presence of purulent bronchial secretion and microbiological examination (mini-BAL [mini bronchoalveolar lavage] or BAL [bronchoalveolar lavage] ≥ 10^4^ CFU/mL, changes in the chest X-ray examination, characteristic auscultatory changes (rales, crackles). VAT was diagnosed on the basis of auscultatory changes (wheezing), purulent discharge from the bronchial tree, microbiological examination as in pneumonia, and no changes in the lungs on chest X-ray [25]. The initial microbiological diagnosis of bronchial secretions was a PCR multitest for 20 respiratory pathogens (FilmARRAY respiratory Panel, BioFire Diagnostics, Salt Lake City, UT, USA). Qualitative and quantitative diagnosis confirming infection with the MIC (Minimal Inhibitory Concentrations) assay was performed in accordance with the standards established by EUCAST (the European Committee on Antimicrobial Susceptibility Testing) [26]. 

### 2.5. Sepsis Diagnosis

The sepsis diagnosis was based on the findings of the Surviving Sepsis Campaign Guideline [27]. 

### 2.6. Data Collection

Data on person days and ventilation days were collected by the epidemiology nurse. Data on compliance with the components of the “VAP-bundle” preventive packages were collected twice a week in the period from October to December 2018 by students from the Students Science Club after appropriate training.

### 2.7. Epidemiological Indicators

***Rate of VAP (or NV-HAP, CAP, VAT)*** *= number of VAP (or NV-HAP, CAP, VAT)/number of patients admitted at ICU at 1 year × 100*

***Incidence Rate*** ***of VAP (or NV-HAP, CAP, VAT**) = number of VAP (or NV-HAP, CAP, VAT)/number patient − days × 1000*

***Incidence density of VAP*** *= number of VAP/number of MV − days × 1000*


***Ventilation utilization ratio V-UR* **
*= number MV-days/total number of patient − days × 100*


### 2.8. Extra Length of Stay and Additional Cost of Therapy

Extra LOS was calculated on the basis of the difference between the average ICU hospitalization time of a patient with VAP or NV-HAP and the mean hospitalization time of a patient without nosocomial infection. The additional cost of therapy resulting from the occurrence of nosocomial VAP or NV-HAP was calculated on the basis of the extra LOS, and the average cost of a person day was calculated by the hospital administration, as seen in the previously published study [19]. The cost of a person day for ICU in 2018 was EUR 732 (3149 PLN). 

### 2.9. Statistical Analysis

The STATISTICA program version 13.1 (StatSoft Inc., Tulsa, OK, USA) was used for statistical analyses in this study. For all study variables, descriptive statistics were computed. Discrete variables were shown as counts and percentages or median and interquartile ranges (IQRs) or 95% confidence interval (CI). The strength of the group distribution of qualitative variables was assessed using chi-square test, the Mann-Whitney U test, or Person’s chi-square test. *p*-value < 0.05 was considered statistically significant.

## 3. Results

Among 578 hospitalized ICU patients (patients’ characteristics are presented in Table 1), RTIs were found in 171 (30%) patients, including pneumonia in 136 (24%). CAP accounted for 34/136 (25%), whereas VAP and NV-HAP (in total) accounted for 102/136 (75%) among the total number of pneumonia cases treated in the ICU. The CAP, NV-HAP, VAP, and VAT rates/100 admissions were 5.88, 9.00, 8.65, and 6.05, respectively. Pneumonia with clinical sepsis was diagnosed in 31/34 (91%), 24/52 (46.15%), and 29/50 (58%) of patients with CAP, NV-HAP, and VAP, respectively. The study found a lower percentage of community-acquired pneumonia 34/578 (5,88%), 95% CI (4.11–8.12) on admission to the ICU than inpatient 102/578 (17.65%), 95% CI (14.62–21.0), *p* = 0. Inpatient pneumonia (on admission and during ICU stays) occurred more often in medical patients 38/196 (19.38%), 95% CI (14.1–25.63%) than in surgical patients 64/382 (16.75%), 95% CI (13.15–20.88%), *p* = 0.4317. The RTIs diagnosed during ICU stays (NV-HAP + VAP + VAT) occurred slightly more often 95/578 (16.4%), 95% CI (13.51–19.71%) than on admission (CAP + NV − HAP) 76/578 (13%), 95% CI (10.34–15.99%), *p* = 0.1336. Hospital-acquired RTIs (NV − HAP + VAP + VAT) occurred statistically significantly more often 137/578 (23.7%), 95% CI (20.29–27.39%) than CAP 34/578 (5.88%), 95% CI (3.82–7.73%) *p* = 0.

Table 2 presents the analysis of the frequency of various clinical forms of RTIs found on admission and during ICU stay.

Table 3 presents the rate of RTIs in terms of gender, age, and reasons for hospitalization. Only CAP and NV-HAP were found statistically significantly more often in medical patients than in surgical patients (*p* = 0 and *p* = 0.006), and VAP was found more frequently in men than in women (*p* = 0.0154).

The incidence rates of CAP/NV-HAP/VAP/VAT/1000 patient-days were 5.83/8.92/8.58/6.0, respectively. The incidence rate of pneumonia during ICU stay (NV-HAP + VAP) was 10.3/1000 patient-days, IQR (8.52–12.32). V-UR at ICU was 79.96%. The mean (IQR) incidence density of VAP/1000 ventilator days over 4660 ventilator days was 10.8 (8.5–12.3).

The analysis of the frequency of VAP in the following months of 2018 is presented graphically in Figure 1.

The most common CAP pathogen was methicillin-sensitive *Staphylococcus aureus* (MSSA), which was responsible for 18/34 (53%) of infections. *Acinetobacter baumannii* was the most common pathogen of nosocomial pulmonary infections. *Acinetobacter baumannii* was multidrug-resistant (MDR) and *Klebsiella pneumoniae* had extended-spectrum beta-lactamase (ESBL)(+) resistance mechanisms that predominated in the NV-HAP/VAP/VAT pathogenesis, accounting for 12/52 (23%) and 33/50 (60%)/15/35, (43%) and 8/52 (15%)/10/50 (20%), and 4/35 (11%) of the total number of these infections, respectively. Gram-negative bacteria with ESBL(+) resistance mechanisms were found in the case of NV-HAP at 11/79 (14%), VAP at 14/79 (19%), and VAT at 7/48 (15%) patients. MDR infections were 12/79 (15%) for NV-HAP and 33/74 (45%) and 16/48 (33%) for VAP and VAT, respectively. The list of RTI pathogens is presented in Table 4.

The mortality among patients diagnosed with CAP, NV-HAP, VAP, and VAT, was 10/34 (29%), 23/52 (44%), 19/50 (38%), and 7/35 (20%), respectively. The mortality in patients with diagnosed VAP was about 11.55% higher than in patients without device-associated healthcare-associated infections (DA-HAIs): 19/50 (38%, 95% CI 24.65–52.83) vs. 107/382 (26.45%, 95% CI 22.08–31.07), *p* = 0.0861). Similarly, a difference in mortality of about 17.55% was observed in the case of NV-HAP: 23/52 (44%, 95% CI 30.47–58.67) vs. 107/382 (26.45%, 95% CI 22.08–31.07) *p* = 0.0082.

The average length of the ICU stay of a patient with diagnosed CAP was 13.9 days, and 13.34 days for those with NV-HAP. Patients with VAP were hospitalized in the ICU for an average of 27.4 days, while patients without nosocomial infection were hospitalized for 9.6 days; *p* = 0.000. The extra LOS caused by VAP/NV-HAP was 17.8/3.7 days, respectively, and resulted in an increase in treatment costs by EUR 13,029/2708 per one infection.

The implementation of preventive packages was assessed on 236 patients. The most frequently observed elements of VAP prevention were the elevation of the head of the bed and rinsing of the mouth with a disinfectant solution, which were found in 94% and 83% of patients, respectively. The least-respected element of VAP prevention was subglottic suction, which was found in 5% of patients. Table 5 presents compliance with elements of VAP prevention.

## 4. Discussion

The results of the study show that respiratory tract infections are a serious epidemiological problem, as they were diagnosed in one-third of patients, and pneumonia was diagnosed in every fourth patient. In the multi-center, international one-day trial European Prevalence of Infection in Intensive Care (EPIC) II covering 75 countries, RTIs were found slightly less frequently in 3091/13,976 (22.4%) patients [13]. In another EPIC III one-day international study covering 88 countries, RTIs were found in 4893 patients among the 15,202 enrolled in the study (32.19%), which is consistent with the results of our study. These infections in the EPIC III study were dominant among all clinical forms of infections (60.1%), which also is in line with the results of previously published studies from our center [19,28,29]. The incidence of RTIs in our study is similar to the results of the multicenter Polish Prevalence of Infections at Intensive Care (PPIC) study, where RTIs were the most common clinical form of infections 70/132 (50.03%) and were found in 70/193 (36.27%) patients treated in ICU [10]. The multicenter European (EU) VAP/CAP study showed that CAP was diagnosed at a lower frequency 262/2436 (10.75%) than nosocomial pneumonia 827/2436 (33.95%), which is similar to our results [5]. The results of our study are also consistent with the results of the EU VAP/CAP study where CAP accounted for 26.5%, whereas VAP and HAP (in total) for 74.0% of the total number of pneumonia patients treated in the ICU (*n* = 136). [5] Our study shows that the incidence of NV-HAP was found more frequently at admission to the ICU from other hospital wards than during ICU stays. It is partly consistent with the results of the Hospital Acquire Pneumonia Prevention Initiative-2 where 70.8% of NV-HAP was diagnosed in hospital wards other than the ICU [6]. A similar incidence rate of pneumonia among patients with HAIs was recorded in consecutive international registers of nosocomial infections conducted by the European Centre for Disease Control (ECDC) [12,16]. Monitoring of lower respiratory tract infections (VAT) in our hospital is a routine procedure; however, this type of infection is not included in international registers of infections such as ECDC, INICC, or the National Healthcare Safety Network (NHSN) regarding monitoring DA-HAIs in the ICU [9,12,16,30]. Published studies indicate that the frequency of VAT occurred in 7–18% of patients treated in the ICU [25,31] (higher than in our study), and the occurrence of this infection prolonged the duration of ventilation and treatment in the ICU [32]. Another promising international study in this area is the ENIRRIs project, which defines, enrolls, and analyzes RTIs from specific epidemiological, clinical, and microbiological perspectives, but the results have not been published yet [24].

Another epidemiological indicator analyzed in the study was the incidence density of VAP and NV-HAP/1000 patient days. The incidence rate of pneumonia (NV-HAP + VAP)/1000 patient days in our study was more than twice as high as in the ECDC registry from 2016 (4.0/1000), IQR (1.0–4.9), and in the same registry from 2017 (3.7/1000), IQR (0.8–4.9) [12,16]. The next epidemiological indicator evaluated in this study was the incidence density of VAP/1000 MV days. The ECDC registers calculate this rate at 1000 intubation days, which may not always coincide with the ventilation days, while the NHSN and INICC registers use the MV days indicator for this purpose [9,11,16,30]. It should be noted that the rate of use of invasive ventilation in our center (VU-R) was 79.96% and was slightly higher than the previously published Polish multicenter study (73.6%) and much higher than in the ICUs participating in the EPIC III study (44.4% for invasive ventilation, 10.3% for non-invasive ventilation) [28,33]. The incidence density of VAP/1000 MV days in this study (10.8), IQR (8.5–12.32) was comparable to the average value from the ECDC register (9.5), although the spread of this indicator in different countries ranged from 2.5/1000 in Luxembourg to 20.4/1000 in Belgium [16]. The same indicator in our study was much lower than the average indicator from Polish ICUs included in the ECDC study, where it was 17.8/1000, which may result from the participation in this study mainly of hospitals with lower referentiality (not university) [12]. Higher than in our center, the above-mentioned index of 12.3 and 15.2/1000 was found in two studies from Polish non-university hospitals [18,34] as well as earlier studies from our center in 2007 and 2010 (16.0/1000 and 18.2/1000) [30,35]. On the other hand, this ratio in our study was lower in comparison with the two low-budget country registries maintained by INICC, where (polled mean VAP rate/1000 or VAE/1000) in internal and surgical wards it was 13.1 (12.9–13.4) 95% CI [11] and 11.13 (10.88–11.38) 95% CI, respectively [36]. A similar relationship was found in relation to the EU VAP/CAP study (18.3/1000) [5]. On the contrary, the rate of VAP/1000 MV days shown in our study is much higher than the results of the Spanish Estudio Nacional de Vigilancia de Infecciones Nosocomiales (ENVIN) registry from 04.2011–12.2012 (9.83–4.34/1000) and the NHSN report from the USA, where in large university centers and ICU wards > 15 beds with an internal and surgical profile, it amounted to 1.6/1000, burns profile 4.4, neurosurgical 2.1, internal medicine 1.0. [9,22]. This may be a result of higher spending on health care in the USA, better organization of epidemiological supervision, or more personnel. This indicator is also higher than the results of the last NHSN register, where the rate of VAE/1000 ventilator days was 6.96 [37].

Our study showed that more than half of patients with RTI presented a clinical picture of sepsis, and it was most often found in patients with CAP and VAP. In a multicenter study Sepsis Occurrence in Acutely Ill Patients (SOAP), the lungs were the most common source of infection found in 68% of patients [38]. On the other hand, in Polish hospitals, the respiratory system was the source of sepsis in 28% of patients in the sepsis register [39].

The analysis found that, regardless of the clinical form of RTI, Gram-negative bacteria (GNB) dominated the pathogenesis of VAP, and the most common pathogen was *Acinetobacter baumannii* MDR. Similarly, it was found in early published studies from our center, where *Acinetobacter baumannii* was responsible for 53.3% of VAP cases and showed an upward trend in 2011–2017 [40]. Differentiation in terms of pathogens of pulmonary infections in different countries is shown in the ECDC registry, where the dominant role in pneumonia in most European hospitals is *Pseudomonas aeruginosa* strain [16], while Gram-positive bacteria (GPB) *MRSA* and *MSSA* dominated in the pathogenesis of VAP and VAT in patients treated in the USA [17,25]. The percentage of multidrug-resistant strains among GNB responsible for RTI in our study is in line with the global trend of MDR infections [17,41].

The study showed that the LOS of patients with RTIs in the ICU was longer (in the case of VAP by 17.8 days, NV-HAP by 3.7 days) than in patients without DA-HAIs and that the treatment costs and mortality were higher in this group of patients. The INICC study found that the occurrence of VAE prolonged hospitalization by an average of 13.5 days and more than doubled the mortality rate in comparison to patients without DA-HAIs (42.2% vs. 17.12%), while in our study, the increase in mortality caused by NV-HAP was 17.55% and was 11.55% in the case of VAP [36]. In a European, multicenter study, it was shown that nosocomial pneumonia prolonged hospitalization in ICU by an average of 12 days. In the same group of patients, RTIs increased mortality by 6% [5]. Another multicenter study showed that VAP increased ICU treatment time by 9 days, hospital treatment time by 13 days, and increased treatment cost by $40,000 USD per VAP [14]. In our study, we also showed that ICU-treated NV-HAP occurred in a similar number of patients as VAP and generated lower treatment costs than VAP but had a slightly higher mortality rate. Slightly different results were obtained in the American study, where NV-HAP was more common than VAP, generated higher treatment costs (which may have resulted from a different cost calculation method), and had a similar mortality rate [42].

The study analyzed the influence of age, gender, and whether a patient had or didn’t have surgery on the incidence of HAP, VAP, and VAT, finding no statistically significant difference in any of the groups. The exception was the more frequent incidence of NV-HAP in internal medicine patients rather than surgical ones, which results from the fact that patients with pneumonia were admitted mainly from non-surgical wards. Additionally, VAP occurred more frequently in surgical patients, which resulted from the ICU profile in which the majority of patients were surgical ones. In a study from Polish hospitals that assessed the risk factors of HAIs, a higher incidence of HAIs was found in men than in women and in patients > 65 years of age compared to in our study [43].

The last element of the study was the assessment of the elements used in VAP prevention [44]. The preventive package includes bed head elevation of 30–50%, use of subglottic suction, maintenance of pressure in the endotracheal/tracheostomy tube balloon > 20 cmH20, use of the sedation protocol with daily awakening or daily evaluation of the possibility of awakening or extubating, use of chlorhexidine for cavity care oral hygiene, maintenance of hygiene of ventilation ducts, use of anticoagulant prophylaxis, and stress ulcers prophylaxis [22,44]. Numerous studies have shown that the implementation of “VAP prevention bundles” may have an impact on the reduction of ventilation time, the frequency of VAP, mortality, and treatment cost [4,19,22,45]. It was also observed that compliance with the individual “bundles” on the level of 70% reduced the number of VAPs from 32/1000 to 12/1000 ventilation-days [45]. The subglottic suction in our study, which was carried out drastically rarely, resulted from the periodic lack of intubation tubes with possibility subglottic suction.

Limitations of the study: Firstly, the study is single-center, so the frequency of respiratory infections as well as the microbiological profile of the ward may be different than in other centers. Secondly, it was not possible to compare all the elements of the study, such as the incidence of particular clinical forms of RTIs in women, men, and general surgery patients and in terms of age due to the lack of published ICU studies in this area. Thirdly, the impact of adherence to bundle elements on the frequency of VAP was not analyzed, as this problem was not included in the assumptions of the study. Fourthly, the economic assessment was guided mainly by the extension of hospitalization time in the ICU and the cost of a person-day; however, the cost calculation and cost of a person day may be different in different centers. Fifth, because we compared our own data with European and USA data on pneumonia monitoring, we would like to underline that other slightly different diagnostic methods for VAP/VAE were used in these studies.

## 5. Conclusions

Infections of the respiratory tract constitute a serious epidemiological problem in patients admitted and treated in ICU, as they affect one-third of patients, and pneumonia affects one-fourth of patients.Hospital-acquired pneumonia was found more frequently than community-acquired pneumonia and had an influence on the length of stay prolongation at the ICU, increasing therapy cost and mortality.Due to the department profile, nosocomial pneumonia VAP was found more often in surgical patients than in internal medicine patients, while NV-HAP was more often found in internal medicine patients admitted from other hospital wards, which requires increased monitoring and the prevention of NV-HAP in internal medicine departments.The frequency and density of VAP occurrence remains high and requires clarification of the cause, tightening the discipline of compliance with preventive packages and elements not included in the study, such as hand hygiene control and an estimation of the number of nursing staff per one patient.

## Figures and Tables

**Figure 1 jcm-11-03764-f001:**
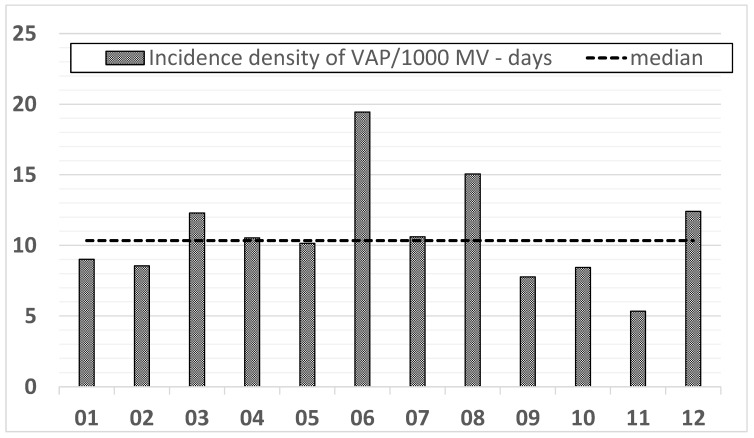
Incidence density of VAP/1000 ventilator days in the following months of 2018.

**Table 1 jcm-11-03764-t001:** Patients’ characteristics.

Year	2018
Total number of hospitalized patients; *n*	578
Women; *n* (%)	219 (37.9)
Men; *n* (%)	359 (62.1)
Surgical patients; *n* (%)	382 (66.1)
Internal medicine patients; *n* (%)	196 (33.9)
Total number of patients-day; (*n*)	5829
Number of patients died; *n* (%)	160 (27.68)

**Table 2 jcm-11-03764-t002:** Frequency of respiratory tract infections during a study period. Data are presented as number of infections and percentage from the total number of hospitalized patients (*n* = 578).

Clinical Kinds of RTI	RTI at Admission at ICU *n* (%)	RTI during Hospital Stay at ICU *n* (%)
CAP	34 (5.88)	-
NV-HAP	42 (7.3)	10 (1.7)
VAP	-	50 (8.65)
VAT	-	35 (6)
RTI total	76 (13.18)	95 (16.4)

Legend: RTI—respiratory tract infections, CAP—community-acquired pneumonia, NV-HAP—non-ventilator hospital-acquired pneumonia, VAP—ventilator-associated pneumonia.

**Table 3 jcm-11-03764-t003:** Rate of RTIs in terms of gender, age, reasons for hospitalization. Data are presented as number, %, 95% CI.

	Number of Patients	CAP *n*, %, (95%CI)	*p*	NV-HAP *n*, %, (95% CI)	*p*	VAP *n*, %, (95% CI)	*p*	VAT *n*, %, (95% CI)	*p*
Surgical patients	382	10, 2.62 (1.26–4.76)	0	26, 6.81 (4.49–9.81)	0.006	35, 9.16 (6.46–12.51)	0.0946	25, 6.54 (4.28–9.51)	0.4913
Internal medicine patients	196	24, 12.24 (8.01–17.67)		26, 13.26 (8.85–18.83)		15, 7.65 (4.35–12,31)		10, 5.1 (2.47–9.18)	
Women	219	14, 6.39 (3.54–10,49)	0.603	17, 7.76 (4.59–12.14)	0.500	11, 5.02 (2.53–8.81)	0.0154	11, 5.02 (2.53–8.81)	0.416
Men	359	20, 5.57 (3.44–8.47)		35, 9.75 (6.89–13.30)		39, 10.86 (7.84–14.55)		24, 6.68 (4.33–9.78)	
≥65 years	255	15, 5.88 (5.49–12.77)	1.0	29, 11.37 (7.75–15.92)	0.076	18, 7.06 (4.24–10.93)	0.2265	13, 5.10 (2.74–8.56)	0.391
<65 years	323	19, 5.88 (3.58–9.03)		23, 7.12 (4.57–10.49)		32, 9.91 (6.88–13.7)		22, 6.81 (4.32–10.13)	

Legend: RTI—respiratory tract infections, CAP—community-acquired pneumonia, NV-HAP—non ventilator hospital-acquired pneumonia, VAP—ventilator-associated pneumonia; *p* was calculated for differences between surgical vs. internal medicine patients, women vs. men, ≥65 years vs. <65 years for CAP, NV-HAP, VAP, and VAT.

**Table 4 jcm-11-03764-t004:** Pathogens of respiratory tract infections. Data was shown as number of pathogens and % from total number of strain responsible for CAP (*n* = 49), NV-HAP (*n* = 79), VAP (*n* = 74) or VAT (*n* = 48).

CAP	NV-HAP	VAP	VAT
*MSSA 18; 37%*	*Acinetobacter baumannii 17; 22%* including *MDR 12; 15%*	*Acinetobacter baumanii MDR 33; 45%*	*Acinetobacter baumannii 20; 42%* including *MDR 15; 31%*
*Escherichia coli 7; 14%*	*Klebsiella pneumoniae/oxytoca 16; 20%* including *ESBL(+) 8; 10%*	*Klebsiella pneum. ESBL+ 10; 14%*	*Klebsiella pneum. 9; 19%* including *ESBL(+) 4; 8%*
*Enterobacter* spp. *5; 10%*	*MSSA 12; 15%*	*MSSA 7; 9%*	*Pseudomonas aeruginosa 4; 8%* including *1; 2% MDR*
*Klebsiella pneum. 4; 6%* including *ESBL(+); 2%*	*MRSA 7; 9%*	*Pseudomonas aeruginosa 6; 8%*	*Enterobacteriacae 4; 8%* including *ESBL(+) 2; 4%*
*Streptococcus pneumoniae 3; 6%*	*Pseudomonas aeruginosa 4; 5%*	*Enterobacteriacae 6; 8%* including *ESBL4; 5%*	*MRSA2; 4%*
*Pseudomonas aeruginosa 2; 4%*	*Enterobacter cloacae* 6; 8% including *ESBL(+)2; 2%*	*MRSA 6; 8%*	*MSSA2; 4%*
*Enterococcus faecalis 2; 4%*	*E coli 5; 6%* including *ESBL(+)1; 1%*	*Enterococcus faecium 1; 1%*	*Serratia marcescens 2; 4%*
Others 8; 16%	Others 12; 15%	Others 5; 7%	Others 6; 12% including *ESBL(+)* 1; 2%

Legends: CAP—community-acquired pneumonia, NV-HAP—non-ventilator hospital-acquired pneumonia, VAP—ventilator-associated pneumonia, VAT—ventilator-associated tracheobronchitis.

**Table 5 jcm-11-03764-t005:** Assessment of the implementation of preventive packages for VAP (*n* = 236 observations). Results are presented as % of follow-up of recommendations from the total number of observations.

Type of Observation	Percentage of Implementation of Recommendations (%)
Raising the head of the bed 30–50%	94.18%
Subglacial suction	5%
Tracheostomy tube balloon pressure >20 cmH20	81.7%
Prevention of stress ulcers	70.79%
Oral rinse with disinfectant	83.24%
Ventilation ducts devoid of bronchial secretions	95.5%
Antithrombotic prophylaxis	84.75%

## Data Availability

The data collected and analysed during this study are available and can be accessed from Wieslawa Duszynska (e-mail: wieslawa.duszynska@umed.wroc.pl).

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
