# Peer review of "Frequency, Etiology, Mortality, Cost, and Prevention of Respiratory Tract Infections—Prospective, One Center Study"

_jcm, 2022, doi:10.3390/jcm11133764_

Round 1

Reviewer 1 Report

The authors presented a prospective study to evaluate the frequency, etiology, mortality, cost, and prevention of Respiratory Tract Infections based on a center in Poland. The study is correct, and the main results and the limitations are discussed. I only have some minor comments to improve the manuscript presentation.

Abstract

Include the objective of the study in the abstract.

“Methods:1. Rate of RTI ; incidence density of VAP / 1000 mechanical ventilator (MV)-days, microbiological profile of RTI, mortality and VAP prevention factors 2. the impact of VAP, NVHAP on the length of stay (LOS) and the additional cost. ”

The methods are written in a very cryptic way; try to rewrite them clearer.

“The extra LOS caused by VAP/NV-HAP was 17.8/3.7 days and increased the cost of therapy by 13,029/2,708 EUR per one patient”.

Define LOS.

“The higher mortality was observed in patients with VAP then with no infections (p = 0.1207).”

Higher mortality compared to what?. Or did you mean the highest?. Also, p is the p-value?. In that case, p-value=0.12 is not significant. The p-value < 0.05 is normally considered significant.

Material and Methods

“Because a statement covering patients data confidentiality was fully respected during data collection and manuscript preparation no consent and statement was needed. ”

This sentence is not clear. No consent was needed?. This issue needs to be clarified.

Change “Ch-square test” to “chi-square test”.

Results

Define “MSSA”.

Change “p = 0.000” to p=0.

“The implementation of preventive packages was assessed on 236 patients. The most frequently observed elements of VAP prevention were the elevation of the head of the bed and rinsing of the mouth with a disinfectant solution, found in 94% and 83% of patients, respectively. The least respected element of VAP prevention was subglottic suction found in 5% of patients. Table 5 presents compliance with elements of VAP prevention”

How did you find any relation between the preventive packages and the different pneumoniaassociated rates you analyzed?. That would be interesting to check.

Discussion

The discussion is long and includes some results that should be moved to the results section, which is shorter. This could be adjusted to balance the manuscript.

“A promising international study in this area is also the ENIRRIs project, the results of which have not been published yet [24]”.

Indicate briefly what the project is about?.

“Care in this area required immediate implementation of recommendations at the nursing level and was improved in the following years (data not shown/published ).”

This sentence is not clear, I suggest to remove it.

Reviewer 2 Report

This paper addresses an important area of clinical practice by examining and describing the incidence of both ventilator associated and non-ventilator RTI within the intensive care setting. The authors have collected a great deal of detailed and useful data across a number of domains. The writing style however makes it a little difficult to follow the flow of the aims, methods and results. I would suggest that the authors seek advice on further editing of the current manuscript.

I feel the work needs further attention.

The underlying methodology for an observational epidemiological study looks robust (clear and accepted case definitions for data items collected and clearly articulated measures of incidence etc.), however the way these methods are presented is hard to follow. I would suggest considering one of the standard study templates for epidemiological research reporting e.g. the STROBE Checklist https://www.strobe-statement.org. Adapting the manuscript to this should not be too onerous for the authors but would greatly help improve the clarity of the paper.

Methods

As a general principle whilst p<0.05 may be considered statistically important it is also important to remember that in observational studies such as this, without pre-determined study power, that a higher p value may hide an important finding.

Results

Please show 95%Cis around proportions, particularly if proportions or rates are comparted to each other or to other studies.

Try and avoid too much repetition of results in tables in again the text - again pull out the key headline results in text. Currently the results are difficult to follow. Perhaps consider the ventilator associated infections first then the non-ventilator associated.

Discussion

Try and pick out the headline messages a little more clearly in the first paragraph. Then go on to discuss in relation to other published work

In the comparisons made between the groups in this study, and in comparisons to other surveys it is important to consider 95Cis around percentages (or proportion estimates).

The apparent high incidence of NV-HAP in other hospital wards is an important observation in that it focuses prevention downstream within the hospital pathway; the authors importantly highlight this in the conclusion. It could be perhaps expanded upon further.

The high rate of VU-R in the study centre when compared to the EPIC III study warrants further discussion; why is this? is it case mix or differences in practice?

The conclusions are important and point to key modifiable risk factors; I would suggest that the paper needs more focus (in effect shortening) to help the reader see how these conclusions are reached.

The shear amount of detailed data on RTI in the ICU settings maybe be very useful to inform future research addressing key questions of modifiable risk; if acceptable under information governance considerations, then publishing of the raw data as supplementary material in anonymised form may help this team and others in this area of work.

The low use of subglottic suction identified by this study is interesting and show the value of simple epidemiological assessment of established risks to improve direct patient care in the hospital setting.
